# Combined Abiotic Stresses: Challenges and Potential for Crop Improvement

Rubab Shabbir [1,2], Rajesh Kumar Singhal [3], Udit Nandan Mishra [4], Jyoti Chauhan [5], Talha Javed [2,6], Sadam Hussain [7], Sachin Kumar [8], Hirdayesh Anuragi [9], Dalpat Lal [10] and Pinghua Chen [1,*]

1 Key Laboratory of Sugarcane Biology and Genetic Breeding, Ministry of Agriculture, National Engineering Research Center for Sugarcane, College of Agriculture, Fujian Agriculture and Forestry University, Fuzhou 350002, China
2 College of Agriculture, Fujian Agriculture and Forestry University, Fuzhou 350002, China
3 ICAR—Indian Grassland and Fodder Research Institute, Jhansi 284003, India
4 Faculty of Agriculture, Sri-Sri University, Cuttack 754006, India
5 Department of Agriculture, School of Science, Noida International University, Noida 203201, India
6 Department of Agronomy, University of Agriculture Faisalabad, Faisalabad 38040, Pakistan
7 College of Agronomy, Northwest A&F University, Yangling 712100, China
8 Centre for Geo-Informatics Research and Training CSK HPKV, Palampur 176061, India
9 ICAR—Central Agroforestry Research Institute, Jhansi 284003, India
10 Department of Agriculture, Jagan Nath University, Chaksu, Jaipur 303901, India
* Correspondence: 000q010027@fafu.edu.cn

**Abstract:** Abiotic stressors are major constraints that affect agricultural plant physio-morphological and biochemical attributes, resulting in a loss of normal functioning and, eventually, a severe decline in crop productivity. The co-occurrence of different abiotic stresses, rather than a specific stress situation, can alter or trigger a wide range of plant responses, such as altered metabolism, stunted growth, and restricted development. Therefore, systematic and rigorous studies are pivotal for understanding the impact of concurrent abiotic stress conditions on crop productivity. In doing so, this review emphasizes the implications and potential mechanisms for controlling/managing combined abiotic stresses, which can then be utilized to identify genotypes with combined stress tolerance. Furthermore, this review focuses on recent biotechnological approaches in deciphering combined stress tolerance in plants. As a result, agronomists, breeders, molecular biologists, and field pathologists will benefit from this literature in assessing the impact of interactions between combined abiotic stresses on crop performance and development of tolerant/resistant cultivars.

**Keywords:** abiotic stresses; omics approaches; molecular breeding; crop improvement; tolerance mechanism

## 1. Introduction

Plants are sessile in nature; therefore, they are frequently subjected to different environmental stresses under field conditions. These stresses include temperature extremes, drought, waterlogging, salinity, and metal toxicity [1]. The occurrence of these stresses at any developmental stage has a drastic impact on plant growth and development processes [2]. It is also well recognized that under field conditions, the plant experiences a combination of these stresses, which pose a more drastic impact than the sole occurrence of these stresses. Salinity and drought [3], salinity and waterlogging [4,5], and temperature and drought stresses [6] frequently coexist under field conditions. The simultaneous occurrence of these stresses endangers crop performance [7].

Salt-alkalization and water deficit stresses obstruct plant homeostasis and ionic balance, resulting in osmotic stress [8]. Under drought stress, salinity causes a significant increase in $Na^+$ and $Cl^-$ concentrations in leaves, which, in turn, can cause a severe reduction in seedling growth [8]. The co-occurrence of salinity and drought stresses increases the

accumulation of phenolics and flavonoids. Under the combined presence of these stresses, a higher drop in water potential leads to cellular component deterioration, restriction in photosynthetic apparatus, stomatal closure, and reduced gas exchange [9].

Excessive soil salinization and waterlogged conditions are also often indistinguishable. Intensive irrigation practices and intensification of saline water tables are the leading causes of combined salinity and waterlogging stress [10]. Salinity stress, when simultaneously occurring with waterlogging stress, can cause considerable damage to crop plants, resulting in a major impact on yield [4]. High accumulation of salts in soil solution reduces the ability of crop plants to uptake soil water, thereby causing drought conditions for plants. The simultaneous incidence of drought, due to uneven rainfall distribution, and temperature stress have resulted in elevated leaf temperature leading to rapid loss of plant water [11]. High temperatures during drought stress resulted in a greater decrease in nutrient uptake and net $CO_2$ diffusion, resulting in reduced photosynthesis [12].

The plants' morpho-physiological and molecular mechanisms to cope with a simultaneous incidence of these abiotic stresses are comparable to some extent [4,5]. Adaptive responses to coupled salt and waterlogging stressors may involve morpho-physiological changes such as the production of adventitious roots, formation of aerenchyma cells, and ion uptake regulation [4]. Under combined salinity and waterlogged stress, reduced accumulations of $Na^+$, $Cl^-$, and $Ca^+$, and more $K^+$ uptake, are also well demonstrated in previous studies [13]. The excessive production of antioxidant enzymes has also been reported for detoxifying reactive oxygen species (ROS) under combined salinity and waterlogging stress [14]. Similar aspects have been reported for combined salinity and drought stresses, as well as heat and drought stresses [15]. At the molecular level, the upregulated expression pattern of the proteins UDP-glucose pyrophosphorylase (UGPase), phosphofructokinase (PFK-B), and alcohol dehydrogenase (ADH), which are involved in glycolysis and fermentation pathways, has been observed under combined salinity and waterlogging stress [16]. Secondary metabolites such as anthocyanins and flavonoids also play essential roles in osmotic adjustment and ROS scavenging under various abiotic stresses. Some stress responsive genes, such as dehydrins, were induced and upregulated for improving tolerance to these stresses [17].

Despite the rising incidence of combined abiotic stresses under field conditions, an astonishing number of research on the solo occurrence of environmental stresses and their independent response in plants has been documented [18,19]. However, plant responses to combined stresses, particularly at physiological and molecular levels, are still not well-understood. Therefore, the current review-based study aimed to evaluate the impact of combined abiotic stresses on plant performance, as well as the adaptive mechanisms in plants, particularly at the physiological and molecular levels. Multiple-omics and other biotechnological approaches to cope with combined abiotic stresses were also discussed.

## 2. Implications of Combined Abiotic Stresses in Plants

Plants are subjected to various sorts of fluctuations in their physical environment. Plant growth is fundamentally dependent on energy (light), water, carbon, and mineral nutrients [20]. Salinity, drought, radiation, high and low temperatures, heavy metals, flooding, and nutrient deficits all have a negative impact on plant growth and development [21]. Insufficient water availability, extreme temperatures, soil nutrients, excess light, and soil salinity and hardening are examples of abiotic stresses that affect a plant's performance [22]. Approximately 50% of the world's cultivated lands are affected by the combination of drought and salinity stress [23]. As a result, plant scientists have become increasingly concerned with understanding the abiotic stress reactions of crop plants for food security.

Plants respond to abiotic stressors in a dynamic and complicated manner, both reversibly and irreversibly [24]. Plant abiotic reactions have been studied physiologically, biochemically, cellularly, and molecularly, revealing intricate cellular responses to abiotic stressors. Furthermore, the intensity and duration of stress, whether acute or chronic, can have a significant impact on the complexity of the response [25]. Abiotic stresses adversely

affect the morphological, biochemical, and physiological processes that are directly related to crop yield and related attributes [22].

### 2.1. Combined Abiotic Stresses and Growth and Development Implications

Drought stress severely influences plants at all stages of growth and development, affecting them from the molecular to morphological levels [26]. Abiotic stressors, such as salt, impact plant roots by increasing osmotic pressure outside the roots [22]. Enzymatic alterations in the cell wall contribute to the inhibition of plant development when water intake and accumulation into growing cells are reduced [24]. Water scarcity restricts plant growth primarily by reducing photosynthesis and respiration. Cold stress disrupted all cellular activities and caused decreased protoplasmic streaming, electrolyte leakage, and plasmolysis, all of which alternately harmed the cells [22]. Salinity, in addition to osmotic stress, creates ionic toxicity, which is linked to nutritional limitations and oxidative damage. High salt concentrations inhibit plant growth and development due to ionic toxicity and osmotic stress [27]. Soil osmotic pressure exceeded plant cell membranes under salt stress conditions, limiting plant uptake of nutrients such as $K^+$ and $Ca^{2+}$ and damaging cell membranes and development. High soil salinity, in addition to inhibiting seed germination, impairs plant growth and development phases due to increased osmotic potential and ion toxicity [2].

### 2.2. Combined Abiotic Stresses and Physiological Implications

Several physiological processes such as photosynthesis, respiration, starch metabolism, and nitrogen fixation are affected under combined abiotic stressors, resulting in crop productivity losses [28]. Lowering the osmotic potential, water potential, and relative water content of leaves, creating nutritional imbalances, and increasing relative stress injury in plants are all examples of physiological changes. Stress physiological reactions also include leaf wilting, leaf abscission, a decrease in leaf area, and decreased water transpiration [29]. Drought stress decreases turgor pressure, which is an important physiological mechanism affecting cell growth [22]. Moreover, drought pressure causes elongated cells, impaired mitosis, and decreased plant height, among other impacts [26]. Drought causes plants to manufacture defensive chemicals by mobilizing metabolites in order to alter their osmotic equilibrium [30]. Osmotic modification can reduce the detrimental effects of stress by maintaining cell water equilibrium [29]. Plant growth and development are reduced when cells and tissues dehydrate and crystallize due to cold stress, which causes water stress and electrolyte leakage, decreased membrane conductivity, and increased water viscosity [31]. Water deficiency inhibits photosynthesis and respiration before it inhibits development. Because of their essential structure, newly split cells around the xylem limit plant growth zones. High temperatures have an impact on plant growth and development, as do irreversible drought stressors that can kill plants [32]. Heat stress causes cells to lose function, resulting in a dysplastic anther during the reproductive growth period [30].

### 2.3. Combined Abiotic Stresses and Molecular Implications

Interactions and crosstalk occur between many molecular pathways during the plant's response to abiotic stresses. Reactive nitrogen species (RNS) and ROS play an important role in many abiotic stress responses, affecting enzyme activity and gene expression [33]. In response to ROS, cells activate several responses, such as increasing the expression of genes responsible for antioxidant functions and producing stress proteins, activating antioxidant enzymes, and accumulating compatible materials [34]. Figure 1 depicts the sensing of ROS by plants based on a hypothetical cascade. Abiotic stress can also be reduced by the presence of certain detoxifying genes such as ascorbate peroxidase, glutathione peroxidase, and glutathione reductase. Abiotic stress responses in plants can also be regulated by hormones (abscisic acid and ethylene) [35]. In many plants, abscisic acid (ABA) plays an important role in regulating osmotic stress. Growth, germination, and protective mechanisms are slower responses to ABA. Cold stress signals are transduced in several ways, such as

ROS components, protein kinase, protein phosphate, ABA, and $Ca^{2+}$, but ABA appears to be the most effective signal transduction pathway [30]. Late embryogenesis abundant proteins (LEA) accumulate in large numbers during early embryogenesis in response to water stress during seed dehydration [22]. In addition to drought and ozone, ethylene is involved in flooding (hypoxia and anoxia), heat, chilling, and ultraviolet light exposure [36]. Figures 2–4 show a schematic illustration of the molecular process for stress endorsement in crop plants.

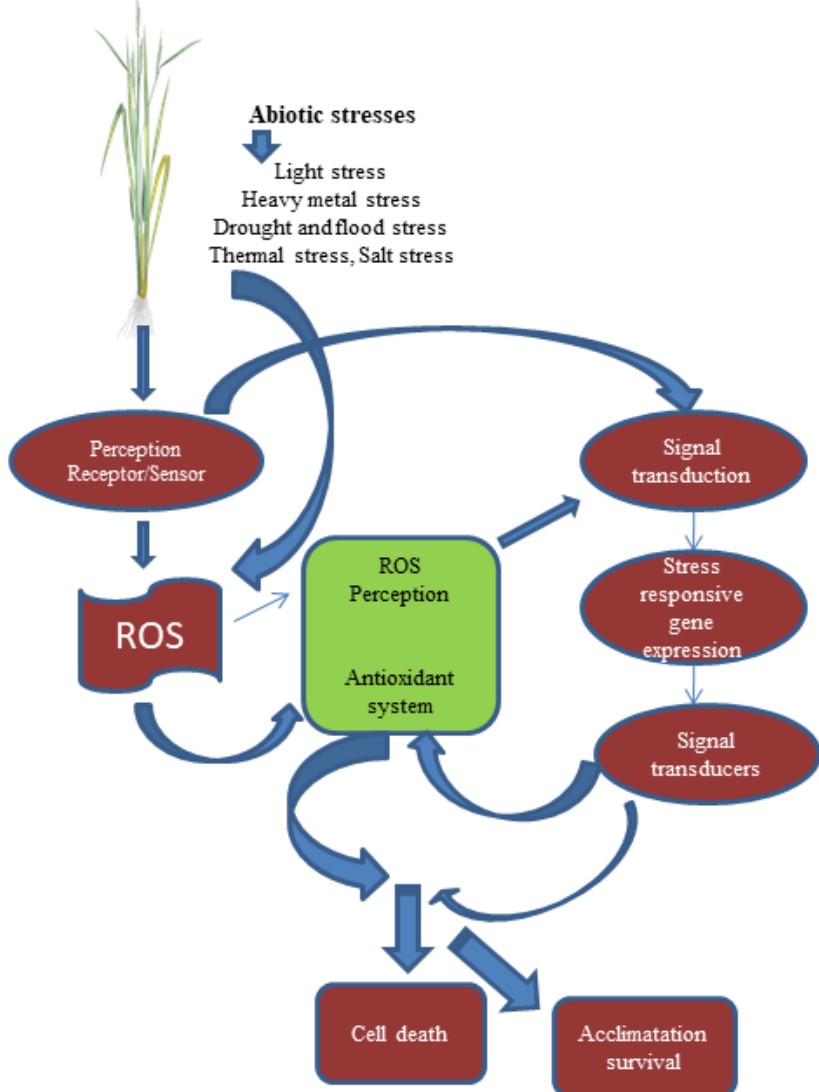

**Figure 1.** Plant perception of ROS based on a hypothetical pathway.

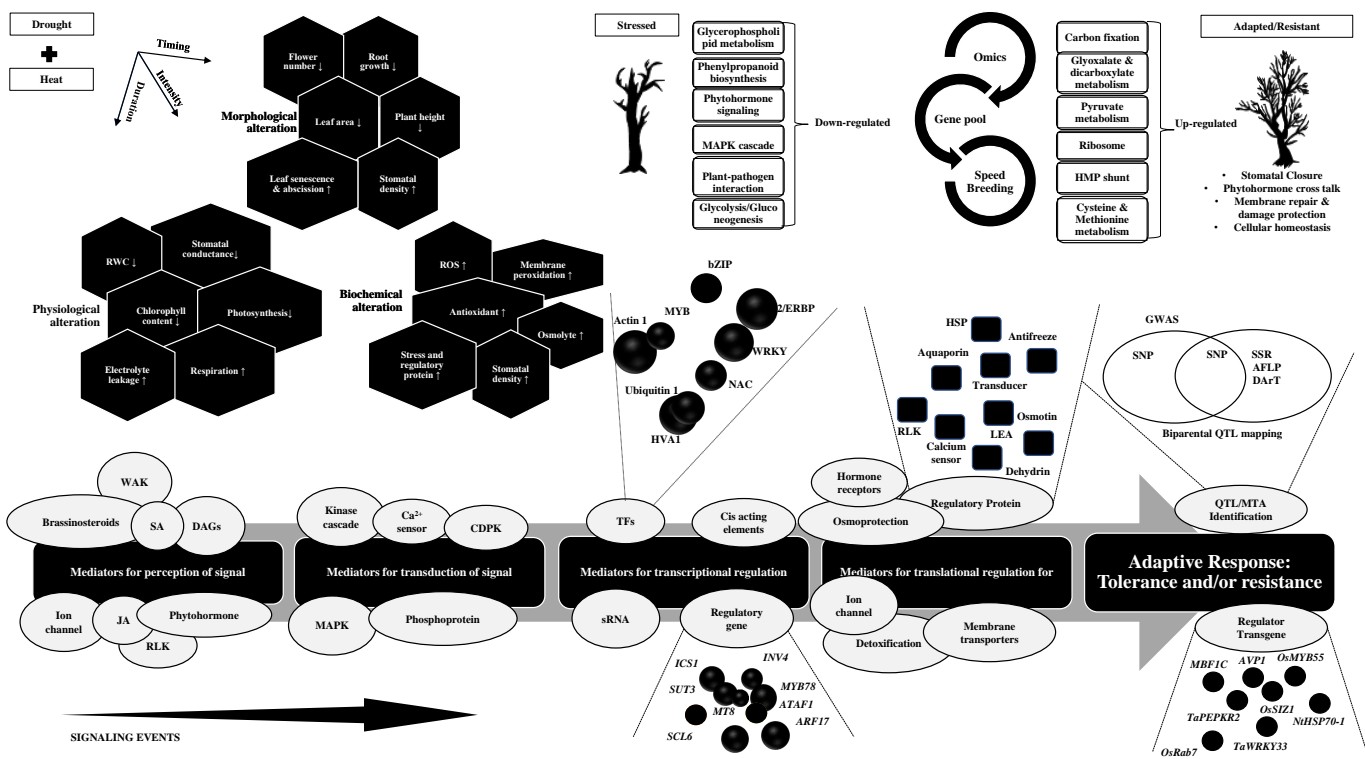

**Figure 2.** Schematic representation of the molecular mechanism for combined drought and heat stress endorsement in crop plants.

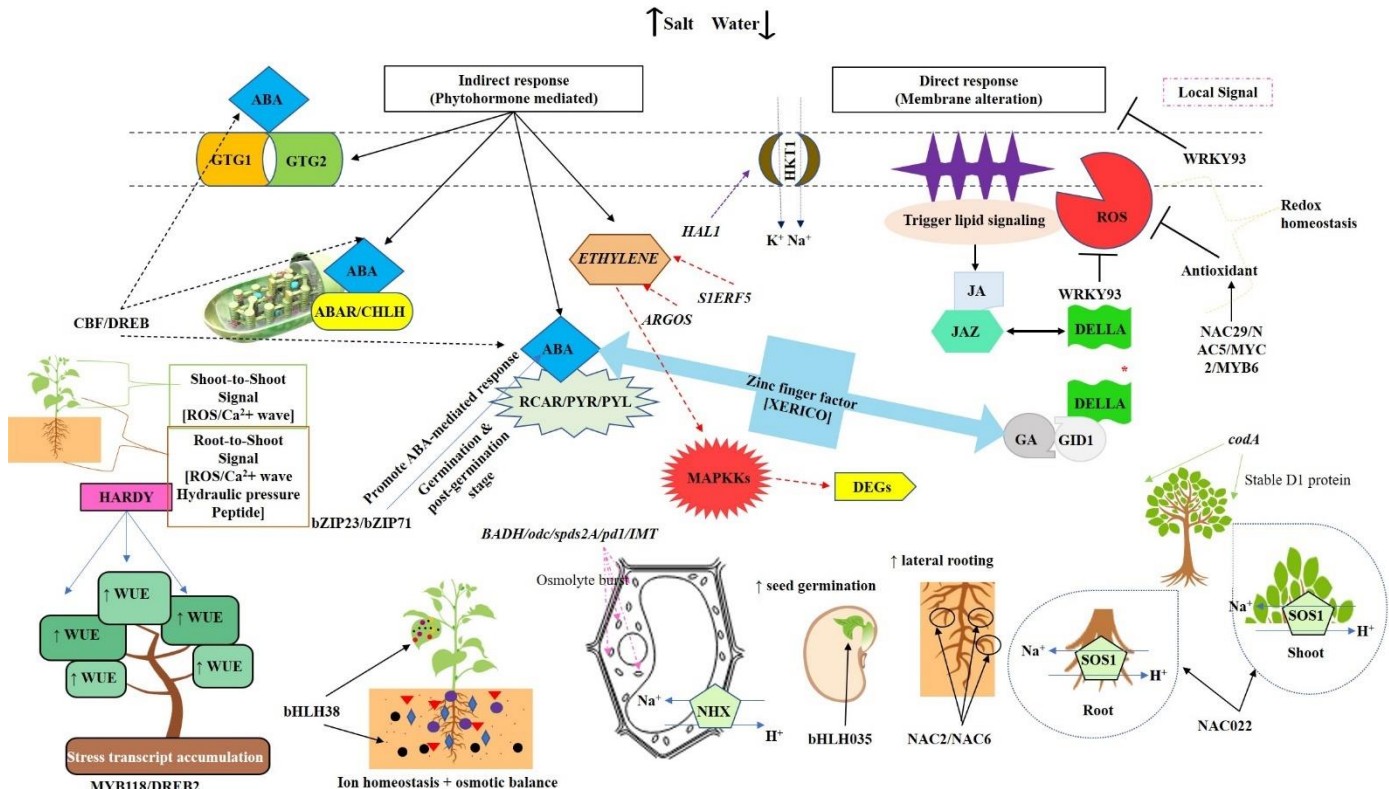

**Figure 3.** Schematic representation of the molecular mechanism for combined drought and salt stress endorsement in crop plants. ↓ & ↑ indicate salt and drought stress, respectively.

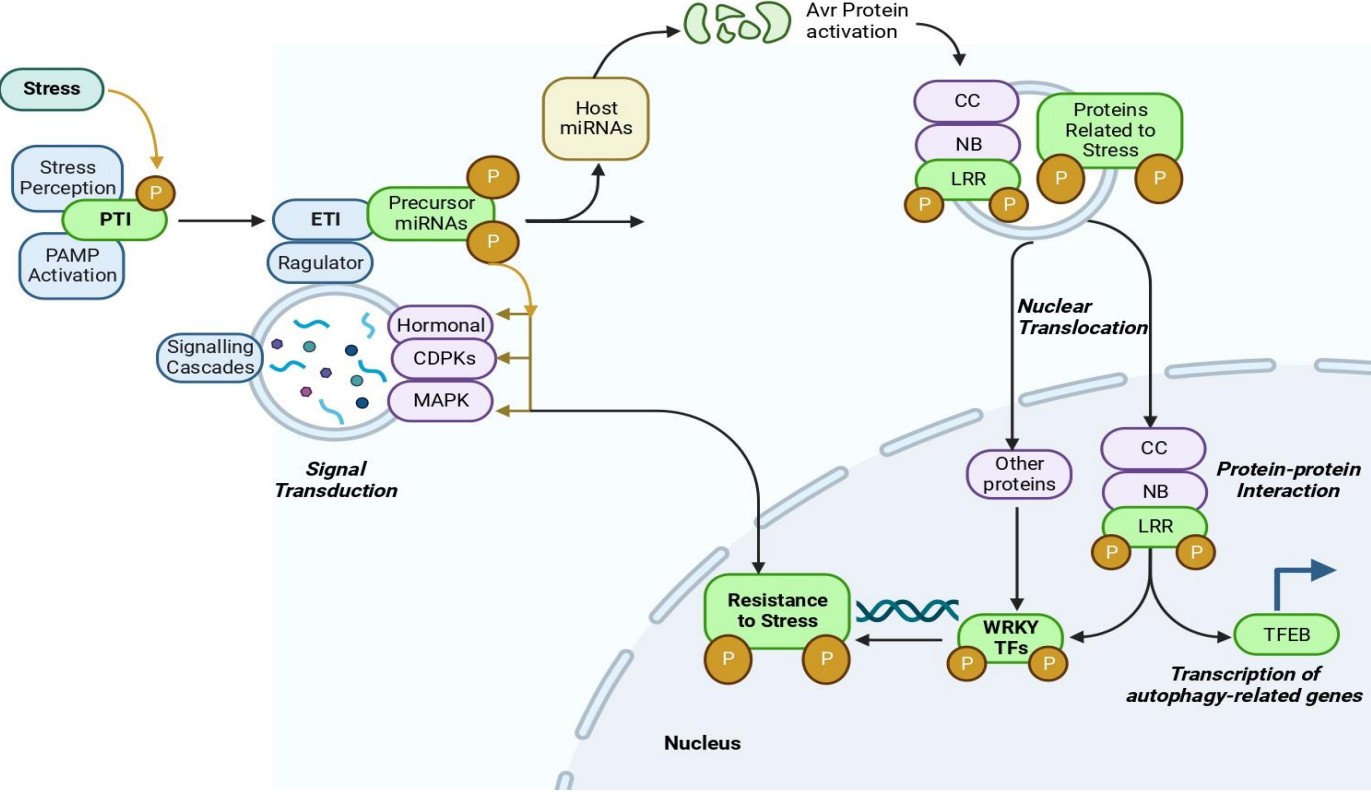

**Figure 4.** Schematic representation of the molecular mechanism for combined abiotic stress endorsement in crop plants.

### 3. Effect of Combined Stress at Different Growth Stages and Crucial Processes

Seed is the prime and most important stage of plant growth and development, and it is responsible for assuring superior progeny for yield and quality attributes in the future. However, seed germination and seedling establishment stages are sensitive to all sorts of stresses under either single or combined forms, with many studies reporting that during the seedling stage combined stresses are more lethal than any form of a single stress. Reduced hypocotyl length (for example up to 10% to 15% in *M. sativa*), poor root hydraulic conductance, root and shoot fresh weight, diminished root activity, disturbance of osmotic balance, impaired metabolic activity at the cellular level, and excessive ROS production, alterations in DNA, RNA and protein structures, membrane damage, reduced respiration, and less ATP production, increased in insoluble carbohydrate contents, were overserved under combined stressors [37–39].

At the vegetative stage, leaf area, plant height, stomatal activity, RWC, and photosynthetic activity are the health indicators for the optimum growth of plants. Simultaneously, combined stress interferes with various physiological processes, resulting in a decrease in their optimal functions. Under heat and salinity combined stress, *Jatropha curcas* stomata were unable to open, and the leaf temperature steadily increased, resulting in a significant reduction in photosynthetic rate and stomatal conductance [40]. Likewise, the combined form of multiple abiotic stressors causes a drop in cellular water content and osmotic potential as well as a decline in growth rate, along with a suite of metabolic changes, stomatal closure, drastic inhibition of photosynthesis and damage of cellular structures, and decreased gas exchange rates [41]. Similarly, photosynthetic gas exchange, chlorophyll fluorescence, nitrogen level, plant biomass, leaf green area, leaf water potential, maximal efficiency of photosystem (PS)II (Fm), and actual PSII efficiency are affected under combined high temperatures and soil drought [42]. Combined herbicide and saline stress enhanced the accumulation of phytohormones (IAA and ABA) and transcription of ethylene, which could be one of the factors responsible for poor salt tolerance in sensitive cultivars [43].

Similarly, Dobra et al. [44] reported that combined heat and drought stress reduced ABA content while increasing auxin concentration in tobacco. Overall, the combined form of stressors causes drastic changes during vegetative growth and is responsible for a decrease in overall plant performance.

The reproductive stage is the most important and sensitive stage to all types of stresses and causes a huge yield penalty. During the reproductive phases, pollination, flowering, grain filling, and grain formation are the most critical stage, which can result in a drastic loss of yield potential. Combined drought and heat stresses affect growth to yield parameters, such as germination, leaf area water use efficiency, nutrient uptake, photosynthetic activity, and grain filling [12,45]. Similarly, in rice floral organs, high expression of the *Carbon Starved Anthers* (CSA) gene leads to sugar starvation in sensitive lines, whereas sugar transporter (*MST8*) and a cell wall invertase (*INV4*) are associated with high sink strength in tolerant lines [46]. Similarly, Rang et al. [47] observed that the number of germinated pollen and stigma fertility in rice are strongly affected under heat and drought combined stresses. Likewise, in canola, heat and drought stress combined reduced the stem pods by 75%, seed pods by 25%, and seed weight by 22%, as compared to the control. Furthermore, the seed yield per plant was reduced by 15% when plants were severely (35/18 °C) stressed during bud formation, 58% when stressed during flowering, and 77% when stressed during pod development.

## 3.1. Antioxidant Defense

ROS are generally produced in plants as a result of multiple abiotic stresses. These ROS act as both signaling molecules and toxins, causing extensive damage to lipid membranes and cellular organelles. Combined stresses led to excessive energetic pressure at the PSII and increased heat dissipation. High antioxidant capacity was genotype-dependent, with higher superoxide dismutase (SOD) and ascorbate peroxidase (APX) activities operating better in the drought-resistant genotype. High SOD and APX activities were associated with a rapid photosynthesis recovery in drought-resistant varieties after drought and low substrate temperature alone or in combination [48]. Similarly, a high accumulation of proline and glycine-betaine was recorded in drought (D) and salt stress (S) tolerant varieties. Moreover, higher levels of ascorbic acid (ASA) and glutathione (GSH), superoxide dismutase (SOD), total ascorbic acid (TAA), $\alpha$-tocopherol, carotenoids, catalase (CAT), glutathione peroxidase (GPX), guaiacol peroxidase (POD), peroxidase (POX), polyphenol oxidase (PPO), non-enzymatic antioxidants, proline, and glutathione reductase (GR) under combined stress were observed with less accumulation of $H_2O_2$ and malondialdehyde [49]. Excess production and the accumulation of ROS cause oxidative damage at the cellular level, disrupt cellular membranes and leads to enzyme inactivation, protein degradation, and ionic imbalance in plants [50]. Furthermore, ROS disrupt the cellular macromolecules including DNA and hence may result in base deletion due to alkylation and oxidation, which are linked to a variety of physiological and biochemical disorders in plants [51].

## 3.2. Mineral Transport Mechanism

Plants require plenty of nutrients to grow well under fluctuating conditions, and 17 macro and micro elements are essential for their optimum function. These minerals participate in energy transduction, signaling, enzymatic reactions, and macromolecule synthesis. Stress, whether single or combined, reduces the absorption, uptake, transport, and efficiency of important elements in the plant system, resulting in abnormal growth and development. Wang et al. [52] used electrophysiological and imaging techniques to investigate the role of Respiratory Burst Oxidase Homolog Protein D (RBOHD) in *Arabidopsis* root responses to combined salinity and hypoxia stress and found the effect on plant homeostasis by poor retention of $K^+$ and reduced uptake of $Na^+$ and $Cl^-$. Likewise, Islam et al. [43] reported that combined herbicide and saline stress lead to the downregulation of $Na^+$ and upregulation of $K^+$ by regulating transporter proteins (*OsHKT1;5*, *OsLti6a,b*,

*OsHKT2;1*, *OsSOS1*, *OsCNGC1*, *OsNHX1*, and *OsAKT1*) in rice and alters the nutrient uptake and transport homeostasis.

*3.3. Signaling Mechanism*

Plants are sessile in nature and coordinate themselves via different signaling mechanisms such as whole plant systematic signaling, redox signaling, ROS signaling, hormonal signaling, and retrograde and anterograde signaling. Abiotic stress, whether single or combined, crosstalk at multiple points at the site of phytohormone synthesis, molecular processes (regulation of the gene, transcription factor, stress protein) disrupt the normal signaling, which ultimately disrupts the normal plant process and functions. Studies showed that combined heat and drought stress produces more ABA than single stress. Furthermore, under these stress conditions, high levels of salicylic acid (SA) antagonize the ABA. Likewise, ROS signaling under multiple stress has an immense role in adapting plants to different stress combinations. Some studies also showed that during combined or single stress, ROS and phytohormone signaling crosstalk with each other in sensing and adapting plants to fluctuating conditions [53,54]. Aside from these, heat, drought, and combined stress significantly altered the phosphorylation levels of 172, 149, and 144 phospho-peptides, respectively, in a maize crop and were linked to multiple abiotic stress tolerance [18]. Further, Zhao et al. [55] found that MAPKs (mitogen-activated protein kinases) act as a cross point in $H_2O_2$ and auxin signaling under Cd and Zn combined stress in rice. Furthermore, they found that MAPKs play a role in the regulation of auxin signaling genes (*OsYUCCA*, *OsPIN*, *OsARF*, and *OsIAA*). Many other novel compounds, such as GABA (Gama amino butyric acid), nitric oxide (NO), mitogen-activated protein kinase kinase 2 (MKK2) (in cold and salt), and Ca play an important role in the regulation of signaling pathways under combined stress [56–58].

## 4. Molecular Mechanism and Signal Transduction Cascades under Combined Abiotic Stresses

Plants often face multiple stresses in a synergistic manner under complex and ever-changing environments, each of which may have antagonistic, synergistic, or additive effects on the plants [59]. For instance, high temperature exhibited antagonistic effects on resistance to tobacco mosaic virus in tobacco [60], whereas salinity stress had synergistic effects on resistance to powdery mildew in barley [61]. Several biomolecules involved in defense mechanisms exhibit intricate and multifaceted interactions that help to improve plant defense systems. To endure multiple stressors, plants activate different signaling cascades and protein-protein interaction networks [2]. Despite the fact that it is a crucial area of research, only a few studies have been conducted to date to understand the plant cellular and molecular mechanisms under combined stresses [62]. With recent advances in biotechnological approaches, such as multiple-omics and Next Generation Sequencing (NGS), elucidating the molecular mechanisms of combined stresses for necessary modification and development of stress resistance in plants has become easier.

Plants under stressful environments produce several biomolecules, such as phytohormones, proteins, etc., which individually or in combination activate a range of metabolic cascades. These phytohormones include ABA, ethylene, jasmonic acid (JA), salicylic acid (SA), etc., which regulate the combined stress signaling pathways. Amongst these, ABA is a major phytohormone. Drought stress induces ABA content, which was found to increase the resistance of tomato against *Botrytis cinerea*. Similarly, salt stress was found to exert enhanced resistance to *Odium neolycopersici* fungus in tomato [63]. In addition, several transcription factors (TFs), ROS, phytohormones, calcium-dependent protein kinases (CDPKs), mitogen-activated protein kinases (MAPK), etc., also significantly regulate signal transduction mechanisms under multiple stresses [64]. In the adaptation mechanism, they involve several kinds of proteins, protein modifiers, adaptors, scaffolds, etc., along with several proteins in an interactive mode. For instance, WRKY TFs were observed to regulate combined biotic and abiotic stresses in several crops through SA, ethylene, and JA, by upregulating

the expression and DNA binding activities upon receiving the stimuli of biotic and abiotic stress [65]. Likewise, NAC is another major TF involved in regulating plant stress. Under combined stress, either overexpression or knockdown has improved the plant defense system in many crops [66]. For example, *Sl*NAC35 was actively involved in the tolerance mechanism under combined drought and pathogen stress [67]. AP2/ERFs are another major TF family actively involved in regulating molecular mechanisms amidst multiple stresses. These include DREB and ERF sub-families. Upregulation of the *PsAP2* exhibited enhanced tolerance to phytohormones' injury in *Papaver somniferum* and multiple stress in transgenic tobacco [68]. Osmotin promoter binding protein 1 (OPBP1) played a major role in inducing disease resistance and salt tolerance. TF families, such as MADS-box, are also involved in regulating combined biotic and abiotic stresses in plants [69]. Further, several transcription factor families, such as inducer of CBF expression (ICE), WRKY, Basic leucine zipper (bZIP), ABRE binding proteins, ABF, AP2/ERF, DREB1, MYB, MYC, HsFs, NACs, Basic/Helix–Loop–Helix (bHLHs), Zinc fingers, etc., are well identified and characterized for an array of abiotic stress tolerance.

Heat-Shock Proteins (HSPs) are among the major proteins involved in a wide range of stresses in plants. These molecular chaperones (10–200 kDa size) mainly protect the plant cell against injury by maintaining the actual conformation of the molecules during stress [70]. Plant cells are also directly protected against multiple stresses by gene encoding products of LEA proteins, osmoprotectants, anti-freeze proteins, detoxification enzymes, and free-radical scavengers. In addition, several endogenous RNA molecules, such as miRNA, have recently been detected to play significant roles in stress responses and tolerance, possibly by cleaving mRNA, repressing translations, remodeling chromatins, or DNA methylation. For instance, Taxak et al. [71] have identified several miRNAs involved in controlling salt and heat stress in combination with *Fusarium* wilt stresses in chickpeas.

Modern system biology approaches involving NGS and omics are effective tools for elucidating the structural and functional aspects at the molecular level under multiple stresses. Genomics and transcriptomics would aid in the identification of useful genes, transcripts, or mechanisms involved in developing tolerant varieties/cultivars against combined stresses. In addition, proteomics approaches would provide comprehensive protein profiling (quantitative and qualitative) under a given set of stresses in order to develop stress resistance in various crops. In conjunction with metabolomics, this would provide a comprehensive insight into the status of specific secondary metabolites, hormones, and signaling molecules generated by plants under a certain set of combined stresses. Proteomics analysis of green alga (*Haematococcus lacustris)* under combined stress of N starvation and high irradiance revealed a total of 49 spots, of them, 13 proteins were downregulated and 36 upregulated and associated with the astaxanthin pathway [72]. Similarly, Rampino et al. [73] reported novel genes that were upregulated in durum wheat under heat stress, drought, and combined stress. These genes were related to stress endorsement, transporter processes, and defense and adaptation processes. Similarly, Alam et al. [16] found 43 proteins, of them 29 were upregulated, 8 were downregulated, and 6 were newly induced and associated with combined salinity and flood stress in maize, and associated with energy, seed storage, secondary metabolite, trafficking, turnover, defense, nitrogen metabolism, and signaling. Similarly, Zhao et al. [74] conducted a proteomics analysis in rice under heat, drought, and combined stress and discovered 201 differentially expressed proteins under combined stress, 16 of which only respond to combined stress, and concluded that these proteins are related to chaperon, protease, and signaling, all of which play important roles in plant adaptation under these conditions. Many other recent studies reveal the molecular mechanism of combined stress and investigate a comprehensive understanding of the mechanism of combined stress.

## 5. Recent Advances in Biotechnological Approaches

Combined drought and temperature stress create secondary stresses, such as oxidative and osmotic stress. These secondary stresses can lead to cell death under extreme circum-

stances [75]. Overall, they activate stress sensors ($Ca^{2+}$), stress receptors (GPCR, RLK, HK, ABA receptor, etc.), transporters (aquaporin, ions, and solute transporters), signal transducers (MAP kinase, CDPKs, etc.), transcription networks (bZIP, MYB AP2/AREB, NAC, HSFs, WRKY TFs, etc.) and chaperones (LEA, HSPs, osmotins, dehydrins, Serum amyloid P; SAP component, etc.) leading to drought and heat stress tolerance [76]. Genome-wide association studies (GWAS) with additional information on SNP markers can unveil the significant traits and their associated markers that govern the tolerance mechanism toward combined drought and heat stress [77]. High throughput genotyping and phenotyping to identify various (quantitative trait loci) QTLs associated with metabolic fates (mQTLs) provide insights into cellular modulations of antioxidant enzymes and their scavenging power for cellular ROS under combined water and/or temperature stress [1,78]. Identification of candidate genes or adaptive loci of genomic regions governing tolerance to combined stress through whole genome resequencing-based GWAS is a promising avenue in the era of plant omics [79]. Transcriptome analysis of stress-exposed genotypes covers several genes which are expressed differentially (differentially expressed genes; DEGs), out of which, genes encoding proline, glycerophospholipid, and oxylipin biosynthetic enzymes, HSPs, and ABA signaling are upregulated and genes associated with $Ca^{2+}$-sensing, photosynthetic proteins, and thiamine biosynthesis, are downregulated [80]. Proteomics study towards the obtainment of proteins that contribute towards plant acclimation to combined heat and water stress deciphers the importance of chaperones, HSPs, derived lipids, and amino acids. Under drought and/or heat stress, the primary tolerance mechanism of plants rests on both syntheses of compatible solutes (differentially expressed metabolites) and altered membrane composition (metabolic reprogramming) [81]. miRNAomics involving miRNA–mRNA duplex recognition, where a single mRNA can be synergistically targeted by multiple miRNAs, can be stated as a strategy of gene regulation in a sequence-specific manner (primed to adaptive strategy) under combined stress. Such regulations occur at one of the many stages of stress signaling, such as miRNA–CDPK and miRNA–RLK at the signal perception stage, miRNA–phytohormone receptor (Auxin response factors; ARF and CBL-interacting protein kinases; CIPK) at the signal relaying stage, with miRNA-TF (HSF, WRKY, myeloblastosis; MYB, and ethylene response factor; ERF) as subsequent downstream effectors [82,83]. Transcriptomics and putative protein interaction network search under combined heat and water stress provide information about the involvement of proteins involved in altered metabolism, redox cellular status, and photosynthesis that are abundantly expressed and have more PPI (protein–protein interaction) for cross-tolerance [84]. Table 1 shows the results of a multi-omics mediated assessment of plant tolerance strategies to various abiotic stress combinations.

**Table 1.** Multi-omics mediated assessment of plant tolerance strategy towards various abiotic stress combinations.

| Sr. No. | Combined Stress Occurrences | Observed Plant Tolerance Strategy | Reference |
|---|---|---|---|
| 1 | Drought + Salinity | ■ $\downarrow$ overall growth, foliage, and biomass<br>■ $\downarrow$ NPR<br>■ Modified RSA ($\downarrow$ root length)<br>■ Altered $K^+/Na^+$ concentration | [85,86] |
| 2 | Drought + Heat | ■ Closure of stomata<br>■ $\downarrow$ photosynthesis<br>■ $\uparrow$ respiration<br>■ Modulation in canopy temperature<br>■ $\uparrow$ cellular aldoses and ketoses<br>■ Altered chlorophyll content<br>■ Differential portioning of metabolites between shoot and root<br>■ Modified RSA ($\downarrow$ root surface area and root mass ratio) | [87–90] |

**Table 1.** *Cont.*

| Sr. No. | Combined Stress Occurrences | Observed Plant Tolerance Strategy | Reference |
|---|---|---|---|
| 3 | Drought + Cold | ■ Altered photosystem-II activity | [48,91] |
| 4 | Drought + Light | ■ ↑ leaf accumulation of anthocyanin<br>■ Altered resonance energy transfer in photosystem to avoid ROS generation through excess $e^-$ excitation | |
| 5 | Drought + Nutrient | ■ ↓ photosynthesis<br>■ ↓ stomatal conductance<br>■ Altered WUE<br>■ ↑ ABA and altered nitrate signaling with respect to low nitrogen status | [92,93] |
| 6 | Drought + ↑ $CO_2$ | ■ ↑ photosynthesis<br>■ Altered dark respiration<br>■ ↑ WUE<br>■ ↓ respiration | [94] |
| 7 | Salt + Heat | ■ ↓ $CO_2$ assimilation<br>■ ↓ efficiency of photosystem-II<br>■ Expression of NDP-kinase 1<br>■ Expression of chlorophyll binding protein<br>■ Expression of ABC transporter I family<br>■ ↑ Cellular concentration of compatible solutes to maintain optimal cellular water potential | [41,95] |
| 8 | Salt + Nutrient | ■ ↑ aquaporin | [96] |
| 9 | Salt + high $CO_2$ or absence of $O_2$ | ■ ↑ biomass with respect to high $CO_2$<br>■ ↑ antioxidants with respect to high $CO_2$<br>■ Altered RSA (↑ root biomass and root elongation) | [97] |
| 10 | Heat + $CO_2$ | ■ ↑ mono- and di-saccharides<br>■ ↑ SOS signaling<br>■ ↑ CAC intermediates<br>■ ↑ GABA shunt | [98–100] |
| 11 | Light + $CO_2$ | ■ ↑ antioxidant capacity | [101] |

↓ & ↑ indicate decreased and increased values, respectively.

High-throughput RNA-seq has hastened the candidate gene expression profiling corresponding to various stresses including salt and/or drought [102]. The discovery and association of elevated and lowered levels of several proteins [103] and metabolites with improved salinity tolerance are mediated by omics study. Additionally, ionomics (elemental profiling) plays a pivotal role in interpreting the relationship between mineral nutrient dynamics (including ion homeostasis) and plant growth stages salinity stress [104]. Under elevated $CO_2$ conditions, both transcriptomics and metabolomics revealed that the outputs of these two omics are in accordance with each other with respect to an increase in the concentration of mono- and disaccharides, organic acid intermediates of the citric acid cycle, aromatic amino acids, flavonoid biosynthesis, and SOS signaling [98,99]. Metabolic profiling of plants under combined high $CO_2$ and temperature depicts important metabolic pathways during which proteins and metabolites were upregulated, including light reaction and TCA cycle, amino acid metabolism, as well as the GABA shunt [100].

The transcriptomics results obtained through high throughput RNA-Seq can be validated by qPCR. Transcriptome and gene ontology annotation of the transcripts unveil the upregulation of transcripts encoding ABA response protein and glyoxylase and the downregulation of transcripts encoding GA and SA as some of the tolerance mechanisms

to combined salt and heat stress. The obtained transcripts from the plants exposed to combined salt and heat stress imply that the phytohormone ABA is required for the acclimation of plants to a combination of salinity and heat stress [54]. Studying the ion profiles (ionomics) revealed differential ion balance ($Na^+/K^+$) with respect to plant species, and the duration of stress exposure as a promising strategy to endure combined salt and temperature stress [41].

In the advent of identifying and screening stress tolerant superior genotypes through phenotypic assessments, phenomics plays a pivotal role. Infrared thermography helps in determining both the phyllospheric temperature [105] and transpiration rate [106] under drought and/or heat stress. Timely stress detection is a key aspect of finding a stress tolerance mechanism. Using spectroscopic variants such as reflectance and Raman spectroscopy, one can be able to perform photosynthetic rate measurements which may come in handy in predicting biochemical events under high light stress [107]. Plant cells in saline environments undergo adaptive mechanisms through either reestablishment of cellular homeostasis, membrane proteins, transporters, and/or modulations in photosynthesis, mineral uptake, and several physio-biochemical activities. In this context, combined genome and transcriptome (TF regulation, $Na^+$ extrusion, and its sequestration, upregulation of osmosensor genes, ROS genes, etc.), proteome (increased PTM plasticity, upregulation in chaperones, NAC, RuBisCO), and metabolome (increased synthesis of compatible solutes, osmoprotectants) studies can confer a strategic advantage to plants in order to impart tolerance to salinity stress alone and in combination with other stressors, such as water scarcity [108]. Metabolomic studies showed cellular accumulation of lipid peroxidase (↑ MDA content) under salt and/or copper exposure stress [109]. Ionomics revealed the accumulation of phosphorous and sodium under salinity as a possible salt tolerance mechanism. Weber et al. [110] reported that the toxic amounts of copper can lead to distortion in nitrogen metabolism by inhibiting nitrate uptake and protein synthesis which may affect the plant's ability to synthesize N-rich osmolytes and hence disturb the indigenous salt tolerance mechanism.

Abiotic stresses such as drought, temperature (high/ low), salinity, cold, ion toxicity, flooding, and ozone are the most threatening stresses and affect the plant growth and development of every crop in the world. In presence of the various abiotic stresses, they decrease by an average of 50% yield loss in the major crops every year globally [111]. Further, the change in climate is also supposed to enhance the strength of abiotic stress and reduce the yield by 20% up to 2050 [112]. To date, many attempts have been made to enhance the against abiotic stress through conventional, molecular breeding, biochemical and biotechnological methods. However, conventional breeding methods and genetic engineering have played a very major role in the development of tolerance against abiotic stress. However, due to the complex nature of gene inheritance and the high interaction between genotype and environment (GXE), these methods were not sufficient to develop a tolerance against abiotic stress [113]. Furthermore, in the last two decades, a lot of work has been undertaken in this direction with the help of genetic engineering techniques to develop transgenic plants for various traits. A cold shock protein (CSP) acts as an RNA chaperone and protects the RNA in stress conditions by a mechanism of binding and unfolding the targeted RNA and maintaining its original condition. The genes (*Csp A* and *Csp B*) of CSP were isolated from *E. Coli* and *Bacillis subtilis*, respectively. Further, *csp A* and *csp B* transformed from bacteria to *Arabidopsis* and the results were shown cold resistance. Furthermore, these genes were also transformed in various crops such as rice, maize, and Arabidopsis, as shown by the tolerance to various abiotic stress like cold, heat, and drought [114]. The transgenic maize "Drought Gard" (Drought tolerant) was developed from the transformation of *cspB* by Monsanto.

The stress tolerance plants developed with genetic engineering have shown promising results, but due to some of the limitations, issues, acceptance, commercialization, and approvals of transgenic crops are slow [115]. To overcome these problems, a recently emerged technique, genome editing, can be an alternative to transgenic, conventional, and

molecular breeding methods to develop stress tolerant plants [116]. These techniques have been successfully implemented to improve various traits and enhance both crop yield and nutritional value [117]. There are three methods of genome editing: Clustered Regularly Interspaced Short Palindromic Repeat/Cas9 (CRISPR/Cas), Transcription Activator-Like Effector Nucleases (TALENs), and Zinc Finger Nucleases (ZFNs) [118]. ZFN and TELEN were expensive, difficult to handle, time-consuming, and resulted in low efficiency and off-target effects, and therefore, are limited in use [64]. However, CRISPR/Cas9 is very popular and the most accepted genome editing tool with high efficiency, precision, ease of handling, and reliable techniques to provide the opportunity for the development of abiotic stress tolerant plants [119]. The CRISPR/Cas9 system is a high-throughput system used in gene transformation, drug delivery, and construction of knockout mutations based on non-homologous end joining (NHEJ) mediated double-stranded break (DSB) [120]. CRISPR/Cas9 has been used in targeting several endogenous genes/transcription factors in various crops, such as Arabidopsis, rice, wheat, maize, lettuce, tomato, potato, and soybean, etc., to develop abiotic stress-resistant plants [121]. Furthermore, several genes have been successfully edited to achieve resistance to multiple stresses *viz.*, drought, ion toxicity, salinity, cold, heat, and nutrient use efficiency (Table 2) [122,123].

**Table 2.** Some of genes targeted by CRISPR/Cas9 tool and their application in abiotic stress tolerance.

| Crop | Targeted Gene | Phenotype/Tolerance | Reference |
|---|---|---|---|
| Arabidopsis | *AtAITRs* | Drought and salt tolerance | [124] |
| | *AtOST2* | Drought tolerance | [125] |
| | *AREB1* | Drought tolerance | [126] |
| Rice | *TIFY1a, TIFY1b* | Cold tolerance | [127–130] |
| | *MYB30* | Cold tolerance | |
| | *SAPK2* | Drought tolerance | |
| | *OsRR22* | Salt tolerance | |
| | *OsEPFL9* | Stomata density regulation | [131] |
| | *OsNRAMP5* | Minimise the cadmium content | [132] |
| | *OsPDS, OsSBEIIb* | Nutrient improvement | [133] |
| | *NRT1.1B* | Enhance nitrogen use efficiency | [134] |
| | *OsEPSPS OsDERF1, OsPMS3 OsMSH1, OsMYB5* | Drought tolerance | [135] |
| | *OsAOX1a OsAOX1b, OsAOX1c OsBEL* | Tolerance against various abiotic stress | [136] |
| | *ABI4, GL1 OST2* | Role in stomata opening | [125] |
| | *OsPRP1* | Cold sensitive | [137] |
| | *OsAnn5* | Cold tolerance | [138] |
| | *OsERA1* | Drought tolerance | [139] |
| | *OsSRL1,2* | Drought tolerance | [140] |
| | *OsVDE* | Salinity tolerance | [141] |
| | *OsDST* | Salinity tolerance | [142] |
| Maize | *ARGOS8* | Drought tolerance and enhance yield | [123] |

**Table 2.** *Cont.*

| Crop | Targeted Gene | Phenotype/Tolerance | Reference |
|------|---------------|---------------------|-----------|
| Wheat | *TaDREB2, TaERF3* | Drought resistance | [143] |
| Lettuce | *NCED4* | Heat tolerance | [144] |
| Tomato | *SlMAPK3* | Heat tolerance | [131] |
| | *SlBZR1* | Heat tolerance | |
| | *SlLBD40* | Drought tolerance | [83] |
| | *SlHyPRP1* | Salinity tolerance | [145] |
| | *SlARF4* | Salinity tolerance | [146] |
| Potato | *Coilin* | Salinity tolerance | [147] |
| Sugarcane | *WRKY49* | Nitrogen stress | [148] |
| Soybean | *GmAITR* | Salinity tolerance | [141] |

## 6. Conclusions and Future Outlook

The presence of abiotic stresses poses a greater drastic impact on crop performance [149–160]. Recent advances in physiology and breeding have disclosed various unknown mechanisms for providing new opportunities for enhancing crop stress tolerance to combined abiotic stresses. Indeed, considerable effort has been made to provide useful information on plants' tolerance mechanisms at the physiological and molecular levels. Higher antioxidant enzymes activities, signal transduction cascades and protein transporters for mineral transport and signaling mechanisms, QTLs associated with metabolic fates, and genotypic interaction with the environment have all been well documented. Under the combined existence of abiotic stresses, the development of tolerant varieties using diversified populations, cost-effective SNP profiling, molecular breeding, QTL mapping, DNA methylation and overexpression/knockdown, as well as CRISPR mediated gnome editing will be the future direction. In order to enhance the performance of tolerant varieties, traditional molecular and breeding approaches should be enhanced greatly by adapting collaborative advances in functional, comparative, and structural genomics. Comparative genomics is expected to provide new insights into developing tolerant varieties with high adaptation to the coexistence of abiotic stresses.

**Author Contributions:** Conceptualization, R.S. and P.C.; writing—original draft preparation, R.K.S., J.C., T.J., S.H., S.K., H.A., U.N.M. and D.L.; writing—review and editing, R.K.S., J.C., T.J. and S.H.; supervision, P.C.; project administration, P.C.; funding acquisition, P.C. All authors have read and agreed to the published version of the manuscript.

**Funding:** This work was supported by the grants from National Natural Science Foundation, P.R. China (31070330, KF2015080, KF2015118). The funders were not involved in the study design, data collection, and analysis as well as data interpretation.

**Data Availability Statement:** Not applicable.

**Conflicts of Interest:** The authors declare no conflict of interest.

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
