# Peer review of "Combined Abiotic Stresses: Challenges and Potential for Crop Improvement"

_agronomy, doi:10.3390/agronomy12112795_

Round 1

Reviewer 1 Report

Dear author,

Congratulations for your work. In the attached file I recommend you some corrections.

Author Response

Dear Reviewer

Thank you for your valuable suggestion and comments for the improvement of current manuscript. We have improved the manuscript based on your worthy comments/suggestions. All the mentioned changes have been incorporated in the manuscript. We appreciate for your warm work earnestly and hope that the correction will meet with approval.

On behalf of all co-authors, once again thank you for your valuable efforts.

Best regards,

Prof. Dr. Pinghua Chen 

Comment-1

Reference number 144 and 145 are the same

Response-1

Esteemed reviewer, we have deleted the reference accordingly. Moreover, we have thoroughly checked and revised the manuscript to prevent any linguistic error.

Reviewer 2 Report

Dear authors, I do see the significance of your manuscript, but please see the below comments to improve it.

Please follow the reference style as required by Journal guidelines and manuscript template. 

Line 42: correct Simultaneously to Simultaneous 

Line 44: correct effect to effects

Line 72: correct combine to combined

Line 124: change 'from morphological levels to molecular' to 'from molecular to morphological levels' in order of magnitude.

Lines 131-321 please rephrase this sentence, in the current form it is unclear. 

Already in this part, many references are excessive, merely repeating something that was already stated.  References should be significantly reduced, even halved. 

Line 216: avoid this type of statement or corroborate them with exact literature - many studies reported during the seedling stage combined stresses are more lethal than any form of a single stress.

Line 245: During the reproductive 'phases:'

Lines 268-275: those abbreviations should be explained when first mentioned. The same applies to all mentioned abbreviations throughout the manuscript. 

The manuscript represents the assembly of existing literature, but I would like to see your opinion on the topic. What is your strategy, and recommendation, where to seek for the genes for resistance, how to combine them, which crops are easier to improve, and which are amongst the most difficult and similar? Do not just list albeit 200 references but please provide your point of view, strategy and directions. 

Author Response

Dear Reviewer

Thank you for your valuable suggestion and comments for the improvement of current manuscript. We have improved the manuscript based on your worthy comments/suggestions. All the mentioned changes have been incorporated in the manuscript. We appreciate for your warm work earnestly and hope that the correction will meet with approval.

On behalf of all co-authors, once again thank you for your valuable efforts.

Best regards,

Prof. Dr. Pinghua Chen

Comment-1

Please follow the reference style as required by Journal guidelines and manuscript template.

Response-1

Thank you for your valuable suggestion. We have checked and revised the reference as per the journal requirement.

Comment-2

Line 42: correct Simultaneously to Simultaneous

Response-2

We have corrected it as per your suggestion.

Comment-3

Line 44: correct effect to effects

Response-3

We have corrected it as per your suggestion.

Comment-4

Line 72: correct combine to combined

Response-4

We have corrected it as per your suggestion.

Comment-5

Line 124: change 'from morphological levels to molecular' to 'from molecular to morphological levels' in order of magnitude.

Response-5

We have corrected it as per your suggestion. Moreover, we have thoroughly checked and revised the manuscript to prevent any linguistic error.

Comment-6

Lines 131-321 please rephrase this sentence; in the current form it is unclear.

Response-6

We have rephrased the sentence and makes its meaning clear as per your suggestion.

Comment-7

Already in this part (line 131-321), many references are excessive, merely repeating something that was already stated. References should be significantly reduced, even halved.

Response-7

Overall, we have revised the manuscript and various references are now updated.

Comment-8

Line 216: avoid this type of statement or corroborate them with exact literature - many studies reported during the seedling stage combined stresses are more lethal than any form of a single stress.

Response-8

The statement has been cited with proper reference

Comment-9

Line 245: During the reproductive 'phases:'

Response-9

We have corrected it as per your suggestion.

Comment-10

Lines 268-275: those abbreviations should be explained when first mentioned. The same applies to all mentioned abbreviations throughout the manuscript.

Response-10

Abbreviations are mentioned explained for the first time wherever necessary followed by their usage subsequently in the text

Comment-11

The manuscript represents the assembly of existing literature, but I would like to see your opinion on the topic. What is your strategy, and recommendation, where to seek for the genes for resistance, how to combine them, which crops are easier to improve, and which are amongst the most difficult and similar? Do not just list albeit 200 references but please provide your point of view, strategy and directions.

Response-11

Esteemed reviewer, thank you for your valuable question. As per your question, we have included one more figure depicting the overall molecular mechanism from stress perception to defense response. We have schematically presented signaling cascades as well as protein-protein interaction networking which play a crucial role in defense response. Future recommendation may include selection of genes based on expression profiling. Those genes may further be validated by different functional studies like Y2H (yeast two hybrid assay) or BiFC assays for determination of interaction networking. Through both methods a complete pathway can be determined and several genes can be linked with each other. We have shown this pathway in a newly added figure 4 where we have presented the protein-protein interaction networking within the nucleus. Secondly polyploidy crops such as sugarcane due to complex genome are very difficult to improve. Extensive improvement has been made throughout the manuscript. Various un-necessary references has been deleted now.

Reviewer 3 Report

A appreciable and need of the hour topic which needs umost attention to address because of rapidly changing climatic conditions. 

My specific observations are 

Line 108 - data is too old, put latest data. It is more than 20 year old. 

123 - heading is morphological implications, but you discussed tissue and cell level biochemistry of the plants. Please discuss here morphological traits of the plants which have been affected or modified by abiotic stresses. 

218 - hypocotyl length  - mention here specific crop with relevant references 

236 - delete fullstop after drought

297 - plant replaced with plants

410 - write fulform of GWAS

General observations 

1. Add some latest references as there are lot of work has been done in various crops

Figures 1 - consider thermal stress which is also very prominent among abiotic stresses during climate change 

Fig. 2 and 3 - Make the figures on whole landscape page so that it will be easily understandable and visualize 

and 2 - 

Author Response

Dear Reviewer

Thank you for your valuable suggestion and comments for the improvement of current manuscript. We have improved the manuscript based on your worthy comments/suggestions. All the mentioned changes have been incorporated in the manuscript. We appreciate for your warm work earnestly and hope that the correction will meet with approval.

On behalf of all co-authors, once again thank you for your valuable efforts.

Best regards,

Prof. Dr. Pinghua Chen

Comment-1

Line 108 - data is too old, put latest data. It is more than 20-year-old.

Response-1

The sentence has been rewritten with suitable reference of recent year

Comment-2

123 - heading is morphological implications, but you discussed tissue and cell level biochemistry of the plants. Please discuss here morphological traits of the plants which have been affected or modified by abiotic stresses.

Response-2

We have changed the heading to address this issue.

Comment-3

218 - hypocotyl length - mention here specific crop with relevant references

Response-3

Dear reviewer, we have added the relevant reference along with specific crop as per your suggestion.

Comment-4

236 - delete full stop after drought

Response-4

Corrected

Comment-5

297 - plant replaced with plants

Response-5

Corrected

Comment-6

410 - write full form of GWAS

Response-6

The full form of GWAS is now mentioned in the manuscript

Comment-7

Add some latest references as there are lot of work has been done in various crops

Response-7

Added now

Comment-8

Figures 1 - consider thermal stress which is also very prominent among abiotic stresses during climate change

Response-8

We have included the thermal stress as per your suggestion.

Comment-9

Fig. 2 and 3 - Make the figures on whole landscape page so that it will be easily understandable and visualize

Response-9

Figure 2 and 3 are now available in full landscape mode with good pixel for better visibility

Round 2

Reviewer 2 Report

Thank you for implementing suggestions and corrections.

Reviewer 3 Report

Fine